# Prognostic impact of catheter ablation in patients with asymptomatic atrial fibrillation

**Tetsuma Kawaji[1,2], Satoshi Shizuta[1]\*, Munekazu Tanaka[1], Shushi Nishiwaki[1], Takanori Aizawa[1], Shintaro Yamagami[3], Akihiro Komasa[1], Takashi Yoshizawa[4], Masashi Kato[2], Takafumi Yokomatsu[2], Shinji Miki[2], Koh Ono[1], Takeshi Kimura[5]**

**1** Department of Cardiovascular Medicine, Graduate School of Medicine, Kyoto University, Kyoto, Japan, **2** Department of Cardiology, Mitsubishi Kyoto Hospital, Kyoto, Japan, **3** Department of Cardiovascular Medicine, Tenri Hospital, Nara, Japan, **4** Department of Cardiology, National Hospital Organization Kyoto Medical Center, Kyoto, Japan, **5** Department of Cardiovascular Medicine, Hirakata Kohsai Hospital, Osaka, Japan

\* shizuta@kuhp.kyoto-u.ac.jp

## Abstract

### Background

Catheter ablation for asymptomatic atrial fibrillation (AF) remains controversial. The aim of the present study was to explore the prognostic impact of catheter ablation in asymptomatic AF patients.

### Methods

We performed a post-hoc analysis of 537 risk-matched pairs of AF patients receiving first-time catheter ablation or conservative management. The primary outcome measure was a composite of cardiovascular death, heart failure (HF) hospitalization, ischemic stroke, or major bleeding. The study patients were divided into asymptomatic and symptomatic patients, and were further divided according to the presence or absence of previous AF-related complications (ischemic stroke or HF hospitalization).

### Results

Most baseline characteristics were well balanced between the catheter ablation versus conservative management groups. The median follow-up period was 5.3 years. Catheter ablation as compared to conservative management was associated with significantly lower incidence of the primary outcome measure in the asymptomatic AF patients (14.7% versus 25.4% at 8-year, log-rank P = 0.008). However, the advantage of catheter ablation was significant only in the high-risk subset of patients with the previous AF-related complications (19.2% versus 55.6% at 8-year, log-rank P = 0.006), but not in those without (13.9% and 17.3%, P = 0.08). On the other hand, among the symptomatic AF patients, catheter ablation was associated with significantly lower incidence of the primary outcome measure regardless of the previous AF-related complications.

**Data Availability Statement:** All relevant data are within the paper.

**Funding:** The authors received no specific funding for this work.

**Competing interests:** The authors declared that no competing interests exist.

**Abbreviations:** AF, atrial fibrillation; CA, catheter ablation; CI, confidence interval; CKD, chronic kidney disease; HF, heart failure; HR, hazard ratio; OAC, oral anticoagulan.

## Conclusions

In the post-hoc analysis of the matched AF cohort, catheter ablation as compared with conservative management was associated with better long-term clinical outcomes among asymptomatic AF patients only when the previous AF-related complications were present.

## Introduction

Atrial fibrillation (AF) is a progressive disease causing cardiac dilatation and heart failure (HF) as well as ischemic stroke. Catheter ablation is a useful rhythm control therapy to reduce AF burden, which is recommended for drug-refractory symptomatic AF in the current guidelines [1,2]. Furthermore, recent studies reported catheter ablation reduced the incidence of subsequent adverse clinical outcomes in symptomatic AF patients with and without concomitant HF [3,4]. Also, several studies have demonstrated favorable outcomes of catheter ablation over conservative management in matched comparisons in general AF population [5–8]. In our previous risk-matched analysis comparing catheter ablation and conservative management for AF in daily clinical practice, ablation was associated with lower risk for a composite of cardiovascular death, HF hospitalization, ischemic stroke or major bleeding. However, the subgroup analysis showed some attenuation of the favorable effect of catheter ablation in patients with asymptomatic AF and those with non-paroxysmal AF [8].

Asymptomatic AF is commonly seen in daily clinical practice and associated with poor prognosis [9,10]. However, scarce data on the clinical outcomes of catheter ablation for asymptomatic AF is available, and its indication remains controversial. Therefore, the aim of this study was to evaluate the clinical impact of catheter ablation in asymptomatic AF patients compared with the conservative management in the previously reported matched AF cohort [8].

## Methods

### Study protocol

The current study was a post-hoc subgroup analysis of the previously reported risk-matched study [8]. Among 4398 patients with diagnosis of AF in Kyoto University Hospital between January 2005 and March 2015, we performed 1:1 matching to find out an appropriate control patient for a given patient in the ablation group with a prespecified method as follows. First, we selected a patient in the ablation group and attempted to find a matched control patient for the selected patient in the ablation group based on the following clinical information; 1) age on the date of first AF documentation (acceptable range: ± 5 years); 2) sex; 3) date of first AF documentation (acceptable range: ± 365 days); 4) types of AF (paroxysmal or chronic); 5) European Hear Rhythm Association (EHRA) symptom grades (1 to 4); and 6) prior history of HF. When we could not find a matched control patient, the selected patient in the ablation group was excluded from the matched analysis. The control patient who had already been chosen was not matched to another patient in the ablation group to ensure 1:1 matching. Finally, we identified 1074 matched patients (537 patients in the ablation group and 537 patients in the conservative group). The detailed method of the matching was described in the previous report [8].

In the present post-hoc subgroup study, we separately assessed the impact of catheter ablation on clinical outcomes in asymptomatic and symptomatic patients with and without previous AF-related complications (ischemic stroke or HF hospitalization).

## Ethics

Follow-up information was obtained by review of hospital-chart and contact by letters and/or phone-call to the patient, relatives, and/or referring physicians. The follow-up protocol in the ablation groups was described in the next section. On the other hand, there were no prespecified follow-up protocol in the conservative group because the present study was a retrospective analysis. The study protocol was approved by the institutional review board in Kyoto University Hospital. Written informed consent for the catheter ablation procedure and follow-up was obtained from all patients in ablation group and we got consent for the enrollment to the study from all patients in the conservative group at the time of follow-up contact.

## Procedural protocol of catheter ablation and post-procedural management

Pulmonary veins isolation (PVI) was performed mostly by radiofrequency catheter ablation using double circular catheters, placing two 20-pollar circular-shaped catheters (Lasso, Biosense Webster or Orbiter PV, C.R. Bard Electrophysiology, Lowell, MA, USA) in ipsilateral superior and inferior pulmonary veins (S1 Table). An 8-mm tip ablation catheter (Fantasista, Japan Lifeline, Tokyo, Japan and NAVISTAER, Biosense Webster, CA, USA) was used from 2004 to 2009, and a 3.5-mm tip irrigation catheter (NAVISTAER THERMOCOOL, Biosense Webster, CA, USA) was used from 2010 to 2015. Cryoballoon (Arctic Front, Medtronic, Inc., MN, USA) was used for PVI only for paroxysmal AF since its introduction to Japan in 2014. Tricuspid valve isthmus ablation was routinely performed regardless of the presence of typical atrial flutter. Superior vena cava was isolated when it was deemed necessary. Complex fractionated atrial electrogram guided ablation was performed when sinus restoration was not obtained after PVI. Additional left atrial linear ablations were performed for sustained atrial tachycardias during the procedure.

A 12-lead electrocardiogram was routinely measured at each clinical visit and 24-hour Holter monitoring was recommended at 3-, 6-, 12-month and yearly thereafter. Antiarrhythmic drug was discontinued before ablation procedure, and was restarted only when recurrent atrial tachyarrhythmias were detected. The second ablation was recommended to patients with recurrent atrial tachyarrhythmias after the blanking period of 3 months. Oral anticoagulant (OAC) was continued for at least 3 months after procedure. Thereafter, discontinuation of OAC in patients without arrhythmia recurrence was left to the discretion of the attending physician.

## Definitions and outcome measures

Symptom status related to AF was classified as Grade 1 to Grade 4 according to the EHRA symptom grade. The EHRA score was proposed as a semi-quantitative measure of AF related symptoms and patients' perception of their general state of health; Grade 1 = none (asymptomatic); Grade 2 = mild/moderate (normal daily activity not affected); Grade 3 = severe (normal daily activity affected); and Grade 4 = disabling (normal daily activity discontinued) [11]. In the present study, asymptomatic AF patients were defined as those with the EHRA Grade 1 symptom status. Symptomatic AF patients were defined with those with the EHRA Grade 2 to 4 symptom status. Previous AF-related complications were defined as hospitalization due to HF and/or ischemic stroke before the index date. AF was classified as paroxysmal (lasting <7 days) or non-paroxysmal (lasting ≥7 days) AF. Low body weight was defined as body weight ≤55 kg in men and ≤50 kg in women.

The primary outcome measure was defined as a composite of cardiovascular death, HF hospitalization, ischemic stroke, or major bleeding. Death was regarded as cardiac in origin unless obvious non-cardiac causes could be identified. Stroke was defined as neurological deficit

requiring hospitalization with symptoms lasting for >24 hours, and categorized into either hemorrhagic or ischemic stroke by computed tomography or magnetic resonance imaging. Major bleeding was defined as International Society of Thrombosis and Hemostasis (ISTH) major bleeding. Secondary outcome measures were the individual components of the primary outcome measure.

The recurrence of atrial tachyarrhythmias was assessed only in the ablation group, which was defined as documented AF and/or atrial tachycardia lasting for >30 seconds or those requiring repeat ablation procedures with a blanking period of 90 days after procedure. Discontinuation of OAC was also assessed only in the ablation group, which was regarded as present when it was intended to be permanent. All clinical outcomes were adjudicated by the study investigators independently from the attending physicians.

### Statistical analysis

Categorical variables were presented as number and percentage and were compared with the chi-square test when appropriate; otherwise, we used Fisher's exact test. Continuous variables were presented as mean and standard deviation or median with interquartile range, and were compared using the Student's t-test or Wilcoxon rank sum test based on their distributions. The cumulative incidence rates of the primary and secondary outcome measures were estimated by the Kaplan-Meier method, and the differences were assessed by the log-rank test. In the ablation group, we also assessed the cumulative incidence of OAC discontinuation and the event-free rate from recurrent atrial tachyarrhythmias with a blanking period of 90 days after the first and the last ablation procedure. Multivariable Cox proportional hazard models were used to estimate the effects of ablation relative to conservative management on the clinical outcome measures, which were expressed as hazard ratios (HRs) with their 95% confidence intervals (CIs). To eliminate the influence of the imbalances in baseline characteristics between the ablation and conservative groups, we constructed multivariable Cox proportional hazard models using those adjusters such as low body weight (<55kg in men and <50kg in women), history of malignancy, chronic kidney disease (CKD) (eGFR ≤60 ml/min/1.73m$^2$) and ischemic stroke. To eliminate the influence of baseline differences in the risks for recurrent atrial tachyarrhythmias, we performed multivariable analysis using the Cox proportional hazard model with 10 covariables (age ≥75 years old, female, low body weight, non-paroxysmal AF, AF interval ≥3 years, hypertension, diabetic mellitus, previous AF-related complications, history of malignancy, CKD). Statistical analyses were performed using JMP 14 pro (SAS Institute Inc, Cary, NC) software. All the analyses were two-tailed, and P value of <0.05 was considered statistically significant.

## Results

### Baseline characteristics

Among the entire 1074 patients (537 pairs), AF symptoms were classified into Grade 1 in 414 patients (38.5%), Grade 2 in 542 patients (50.5%), Grade 3 in 104 patients (9.7%), and Grade 4 in 14 patients (1.3%) (Fig 1). Asymptomatic AF patients (Grade 1) were younger and had significantly higher prevalence of male, non-low body weight, non-paroxysmal AF, previous AF-related complications, CKD, low left ventricular ejection fraction, and left atrial dilatation compared with symptomatic AF patients (S2 Table). Most baseline characteristics were well balanced between the ablation and conservative groups in asymptomatic AF patients, except for the prevalence of low body weight, history of malignancy, and previous ischemic stroke (Table 1).

The prevalence of previous AF-related complications was significantly higher in asymptomatic AF patients compared to symptomatic AF patients (23.4% vs. 11.5% P<0.001). Patients with the previous AF-related complications had significantly higher cardiovascular risks and

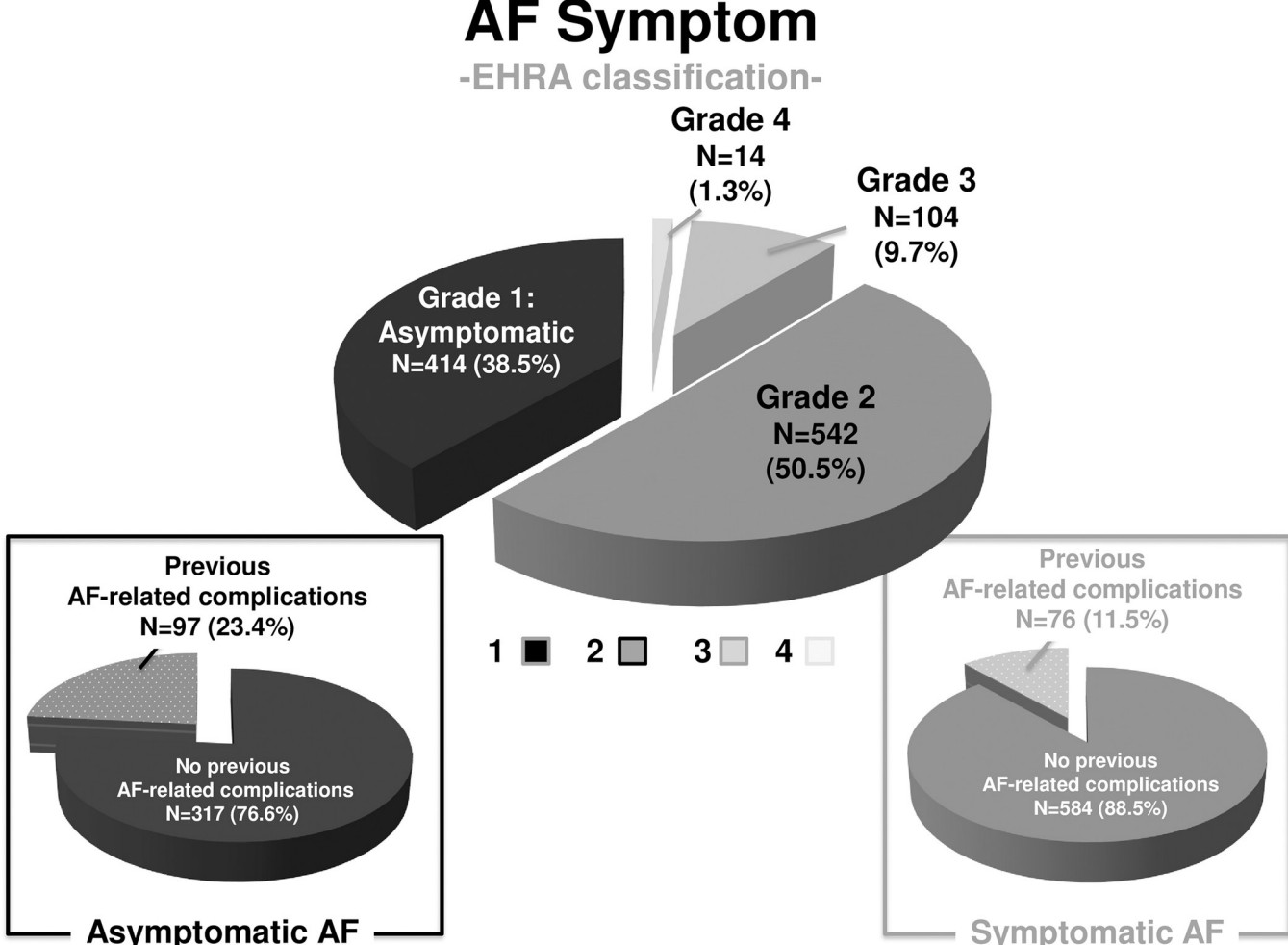

**Fig 1. EHRA symptom grades and the prevalence of previous AF-related complications.** AF = atrial fibrillation; EHRA = European Hear Rhythm Association.

lower cardiac function than those without, regardless of AF symptoms (S3 Table). Most baseline characteristics in asymptomatic AF patients with and without previous AF-related complications were also well balanced between the ablation group and conservative group, except for the prevalence of low body weight, history of malignancy, low left ventricular ejection fraction and CKD (S4 Table).

### Clinical outcome measures compared between ablation group and conservative group

The median follow-up duration was 5.3 (3.8–7.0) years. The 8-year cumulative incidence of the primary outcome measure was significantly lower in the ablation group than in the conservative group, both in the asymptomatic and symptomatic AF patients (14.7% versus 25.4%, log-rank P = 0.008; 7.7% versus 25.8%, log-rank P<0.001) (Fig 2). Even after adjustment of the imbalances in the baseline characteristics, the lower risk of the ablation group relative to the conservative group for the primary outcome measure remained highly significant, both in the asymptomatic and symptomatic AF patients (adjusted HR 0.46, 95% CI 0.25–0.82, P = 0.009; adjusted HR 0.20, 95% CI 0.11–0.35, P<0.001).

**Table 1. Baseline characteristics comparing conservative versus ablation groups in asymptomatic and symptomatic AF patients.**

| Baseline characteristics | Asymptomatic (N = 414) | | | Symptomatic (N = 660) | | |
|---|---|---|---|---|---|---|
| | Conservative group N = 207 | Ablation group N = 207 | P value | Conservative group N = 330 | Ablation group N = 330 | P value |
| Age (years old) | 66.3±6.9 | 66.0±6.6 | 0.70 | 66.5±9.0 | 66.3±8.5 | 0.75 |
| $\geq$ 75 years old | 21 (10.1%) | 20 (9.7%) | 0.87 | 57 (17.3%) | 58 (17.6%) | 0.92 |
| Women | 32 (15.5%) | 32 (15.5%) | 1.00 | 112 (33.9%) | 112 (33.9%) | 1.00 |
| Weight (kg) | 63.1±12.1 | 66.6±12.1 | 0.005 | 60.0±12.1 | 63.5±13.0 | <0.001 |
| Low body weight | 44 (23.7%) | 27 (13.0%) | 0.006 | 83 (28.3%) | 65 (19.7%) | 0.01 |
| Non-paroxysmal AF | 116 (56.0%) | 116 (56.0%) | 1.00 | 52 (15.8%) | 52 (15.8%) | 1.00 |
| AF duration (years) | 1.3 (0.4–4.5) | 1.3 (0.4–4.5) | 1.00 | 1.4 (0.4–3.8) | 1.4 (0.4–3.8) | 1.00 |
| Hypertension | 134 (64.7%) | 149 (72.0%) | 0.11 | 207 (62.7%) | 202 (61.2%) | 0.69 |
| Diabetes | 49 (23.7%) | 34 (16.4%) | 0.07 | 62 (18.8%) | 53 (16.1%) | 0.36 |
| Previous AF-related complications | 42 (20.3%) | 55 (26.5%) | 0.13 | 42 (12.7%) | 34 (10.3%) | 0.33 |
| History of heart failure hospitalization | 19 (9.2%) | 18 (8.7%) | 0.86 | 12 (3.6%) | 9 (2.7%) | 0.51 |
| Ischemic stroke | 25 (12.1%) | 42 (20.3%) | 0.02 | 32 (9.7%) | 28 (8.5%) | 0.59 |
| $CHA_2DS_2$-VASc score | 2.2±1.5 | 2.3±1.5 | 0.44 | 2.3±1.5 | 2.2±1.4 | 0.36 |
| $\geq$ 2 | 133 (64.3%) | 134 (64.7%) | 0.92 | 219 (66.4%) | 200 (66.7%) | 0.93 |
| History of malignancy | 53 (25.6%) | 20 (9.7%) | <0.001 | 85 (25.8%) | 25 (7.6%) | <0.001 |
| eGFR (ml/min/1.73m$^2$) | 54.1±15.3 | 52.1±18.4 | 0.06 | 52.9±18.5 | 55.6±15.8 | 0.046 |
| $\leq$60 ml/min/1.73m$^2$ | 391 (73.0%) | 337 (67.4%) | 0.051 | 197 (64.8%) | 222 (67.5%) | 0.048 |
| **Echocardiographic data** | | | | | | |
| Left ventricular ejection fraction (%) | 64.2±10.8 | 65.1±11.9 | 0.22 | 65.7±9.8 | 66.7±10.9 | 0.27 |
| $\leq$ 40% | 27 (5.0%) | 16 (3.4%) | 0.21 | 6 (2.0%) | 7 (2.1%) | 0.93 |
| Left atrial diameter (mm) | 40.8±6.5 | 41.2±9.2 | 0.54 | 39.4±8.5 | 39.7±6.0 | 0.51 |
| $\geq$ 50 mm | 56 (10.5%) | 72 (15.6%) | 0.02 | 35 (11.9%) | 25 (7.6%) | 0.07 |
| **Medications** | | | | | | |
| Oral anticoagulants | 169 (81.6%) | 207 (100%) | <0.001 | 219 (66.4%) | 330 (100%) | <0.001 |
| Warfarin | 104 (50.2%) | 104 (50.2%) | 1.00 | 109 (33.0%) | 175 (53.0%) | <0.001 |
| DOACs | 65 (31.4%) | 103 (49.8%) | <0.001 | 112 (33.9%) | 155 (47.0%) | <0.001 |
| Antiplatelets | 50 (24.2%) | 43 (20.8%) | 0.41 | 75 (22.7%) | 55 (16.7%) | 0.049 |
| Anti-arrhythmic drugs | 31 (15.0%) | 45 (21.7%) | 0.07 | 120 (36.4%) | 118 (35.8%) | 0.87 |
| Beta blockers | 82 (39.6%) | 63 (30.4%) | 0.05 | 107 (32.4%) | 107 (32.4%) | 1.00 |
| Verapamil/diltiazem | 34 (16.4%) | 18 (8.4%) | 0.02 | 67 (20.3%) | 38 (11.5%) | 0.002 |
| Digitalis | 30 (14.5%) | 21 (10.1%) | 0.18 | 32 (9.7%) | 26 (7.9%) | 0.41 |
| ACEI/ARB | 91 (44.0%) | 100 (48.3%) | 0.37 | 117 (35.5%) | 138 (41.8%) | 0.09 |

Categorical variables are presented as number (percentage). Continuous variables are presented as mean ± SD or median and interquartile range.

AF = atrial fibrillation; BNP = brain natriuretic peptide; DOACs = direct oral anticoagulants; eGFR = estimated glomerular filtration rate; EHRA = European Heart Rhythm Association.

Among the asymptomatic AF patients, catheter ablation was associated with significantly lower risk of the primary outcome measure when the previous AF-related complications were present (19.2% versus 55.6% at 8-year, log-rank P = 0.006; adjusted HR 0.39, 95% CI 0.17–0.90, P = 0.03), but not in the absence of the previous AF-related complications (13.9% versus 17.3% at 8-year, log-rank P = 0.08; adjusted HR 0.51, 95% CI 0.22–1.17, P = 0.11) (Fig 3, Table 2). However, among the symptomatic AF patients, the advantage of catheter ablation was prominent regardless of the presence or absence of the previous AF-related complications.

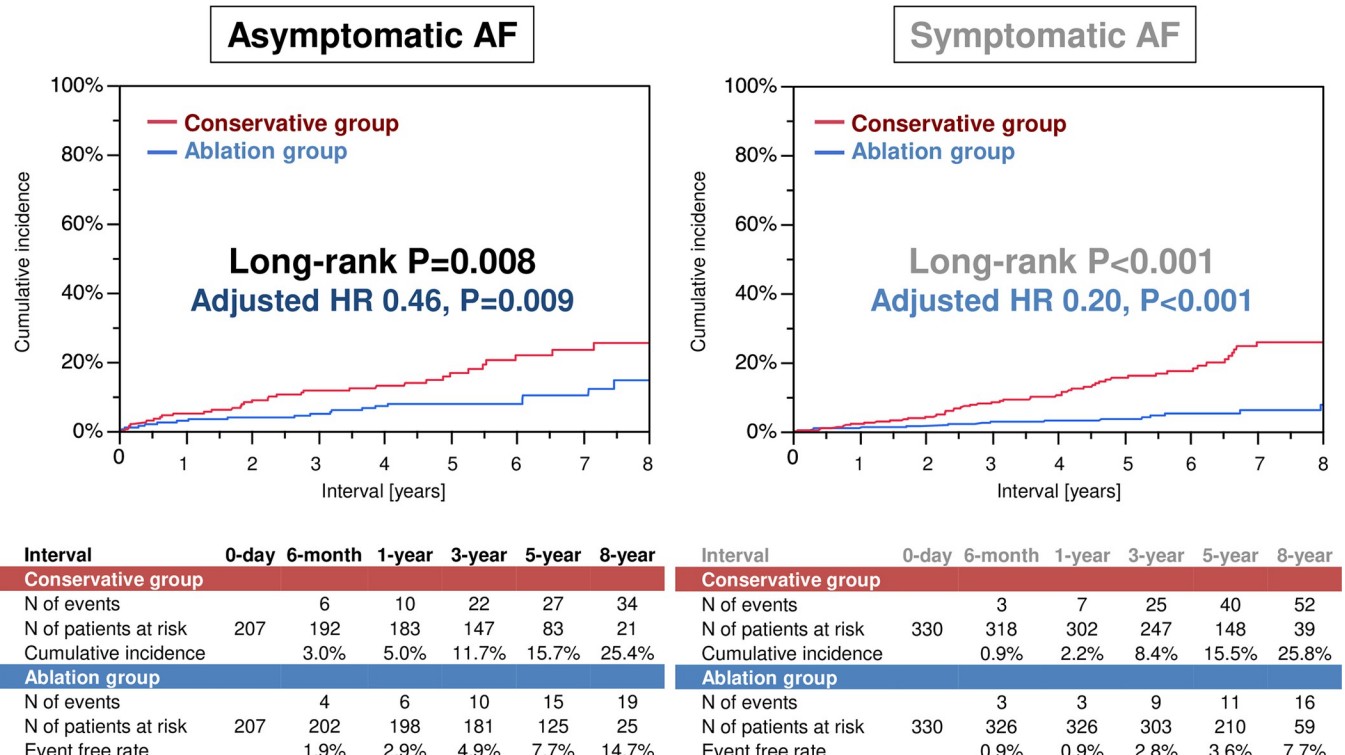

The following table is part of Fig 2.

| Interval | 0-day | 6-month | 1-year | 3-year | 5-year | 8-year |
|---|---|---|---|---|---|---|
| **Conservative group** | | | | | | |
| N of events | | 6 | 10 | 22 | 27 | 34 |
| N of patients at risk | 207 | 192 | 183 | 147 | 83 | 21 |
| Cumulative incidence | | 3.0% | 5.0% | 11.7% | 15.7% | 25.4% |
| **Ablation group** | | | | | | |
| N of events | | 4 | 6 | 10 | 15 | 19 |
| N of patients at risk | 207 | 202 | 198 | 181 | 125 | 25 |
| Event free rate | | 1.9% | 2.9% | 4.9% | 7.7% | 14.7% |

| Interval | 0-day | 6-month | 1-year | 3-year | 5-year | 8-year |
|---|---|---|---|---|---|---|
| **Conservative group** | | | | | | |
| N of events | | 3 | 7 | 25 | 40 | 52 |
| N of patients at risk | 330 | 318 | 302 | 247 | 148 | 39 |
| Cumulative incidence | | 0.9% | 2.2% | 8.4% | 15.5% | 25.8% |
| **Ablation group** | | | | | | |
| N of events | | 3 | 3 | 9 | 11 | 16 |
| N of patients at risk | 330 | 326 | 326 | 303 | 210 | 59 |
| Event free rate | | 0.9% | 0.9% | 2.8% | 3.6% | 7.7% |

**Fig 2. The Kaplan-Meier curves for the cumulative incidence of the primary outcome measure defined as a composite of cardiovascular death, heart failure hospitalization, ischemic stroke, or major bleeding in asymptomatic and symptomatic AF patients.** AF = atrial fibrillation; HR = hazard ratio.

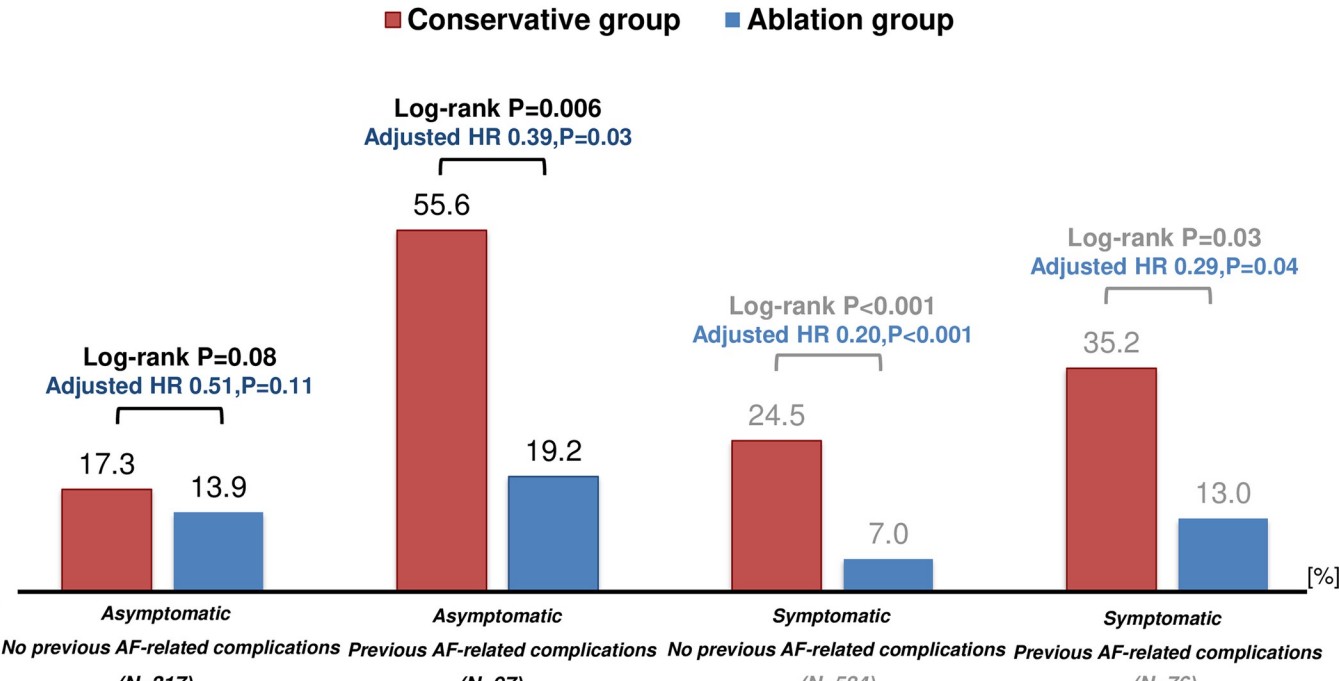

**Fig 3. The 8-year cumulative incidence of a composite of cardiovascular death, heart failure hospitalization, ischemic stroke, or major bleeding according to the symptom status and previous AF-related complications.** HR = hazard ratio; LVEF = left ventricular ejection fraction.

**Table 2. Clinical outcomes: Ablation versus conservative management in asymptomatic and symptomatic AF patients with and without previous AF-related complications.**

**A) Asymptomatic AF**

| Asymptomatic AF | No previous AF-related complications | | | | | Previous AF-related complications | | | | |
|---|---|---|---|---|---|---|---|---|---|---|
| Clinical outcomes | Conservative group N = 162 N of patients with event (Cumulative 8-year incidence) | Ablation group N = 152 N of patients with event (Cumulative 8-year incidence) | Adjusted | | | Conservative group N = 42 N of patients with event (Cumulative 8-year incidence) | Ablation group N = 55 N of patients with event (Cumulative 8-year incidence) | Adjusted | | |
| | | | HR | 95% CI | P value | | | HR | 95% CI | P value |
| Primary outcome measure | 18 (17.3%) | 10 (13.9%) | 0.51 | 0.22–1.17 | 0.11 | 16 (55.6%) | 9 (19.2%) | 0.39 | 0.17–0.90 | 0.03 |
| Secondary outcome measures | | | | | | | | | | |
| Cardiovascular death | 7 (5.9%) | 3 (7.1%) | 0.36 | 0.07–1.44 | 0.15 | 7 (25.1%) | 4 (9.3%) | 0.68 | 0.18–2.55 | 0.55 |
| Heart failure hospitalization | 7 (5.8%) | 3 (4.3%) | 0.40 | 0.08–1.56 | 0.19 | 8 (27.9%) | 5 (10.2%) | 0.45 | 0.17–1.73 | 0.30 |
| Ischemic stroke | 3 (3.4%) | 3 (6.0%) | 1.38 | 0.21–12.1 | 0.74 | 4 (14.4%) | 0 (0.0%) | - | - | 0.003 |
| Major bleeding | 5 (6.5%) | 2 (1.4%) | 0.38 | 0.05–2.22 | 0.29 | 2 (16.5%) | 2 (4.4%) | 0.49 | 0.06–4.18 | 0.49 |

**B) Symptomatic AF**

| Symptomatic AF | No previous AF-related complications | | | | | Previous AF-related complications | | | | |
|---|---|---|---|---|---|---|---|---|---|---|
| Clinical outcomes | Conservative group N = 288 N of patients with event (Cumulative 8-year incidence) | Ablation group N = 296 N of patients with event (Cumulative 8-year incidence) | Adjusted | | | Conservative group N = 42 N of patients with event (Cumulative 8-year incidence) | Ablation group N = 34 N of patients with event (Cumulative 8-year incidence) | Adjusted | | |
| | | | HR | 95% CI | P value | | | HR | 95% CI | P value |
| Primary outcome measure: | 42 (24.5%) | 13 (7.0%) | 0.20 | 0.10–0.37 | <0.001 | 10 (35.2%) | 3 (13.0%) | 0.29 | 0.06–0.95 | 0.04 |
| Secondary outcome measures | | | | | | | | | | |
| Cardiovascular death | 7 (4.4%) | 1 (0.4%) | 0.08 | 0.004–0.48 | 0.003 | 3 (10.5%) | 1 (3.9%) | 0.62 | 0.03–4.93 | 0.67 |
| Heart failure hospitalization | 17 (8.7%) | 7 (3.1%) | 0.35 | 0.13–0.88 | 0.03 | 7 (26.3%) | 2 (9.5%) | 0.33 | 0.05–1.42 | 0.14 |
| Ischemic stroke | 13 (6.7%) | 4 (3.2%) | 0.22 | 0.06–0.64 | 0.005 | 1 (2.5%) | 1 (3.2%) | 0.52 | 0.02–6.28 | 0.60 |
| Major bleeding | 12 (9.8%) | 3 (1.3%) | 0.12 | 0.03–0.38 | <0.001 | 2 (5.5%) | 0 (0.0%) | - | - | 0.050 |

The primary outcome measure was a composite of all the secondary endpoints. The number of patients with event was calculated throughout follow-up period, and cumulative incidence was censored at 8-year.

AF = atrial fibrillation; CI = confidence interval; HR = hazard ratio.

The secondary outcome measures compared between the ablation and the conservative groups are shown in Table 2. The event rates were at least numerically lower in the ablation group than in the conservative group in all the subgroups, except for ischemic stroke in asymptomatic AF patients without previous AF-related complications.

## Recurrence of atrial tachyarrhythmias and discontinuation of OAC in the ablation group

The event-free rate from recurrent atrial tachyarrhythmias after the first ablation procedure was not significantly different between asymptomatic and symptomatic AF patients (54.5%

and 61.2% at 8-year, log-rank P = 0.05), while the arrhythmia-free rate after the last procedure was significantly lower in the asymptomatic AF patients (77.6% and 88.5% at 8-year, log-rank P = 0.002) (S1A Fig). However, the adjusted risks for the recurrent atrial tachyarrhythmias after the first and the last ablation procedures were not significant different between asymptomatic and symptomatic patients (adjusted HR 0.95, 95% CI 0.69–1.31, P = 0.77; adjusted HR 1.23, 95% CI 0.73–2.07, P = 0.43). Furthermore, the arrhythmia-free rates after the first and the last ablation procedures were not significantly different between asymptomatic AF patients with and without the previous AF-related complications (42.9% versus 58.8%, log-rank P = 0.11; 72.9% versus 78.8%, log-rank P = 0.21) (S2A Fig).

The prevalence OAC at baseline was 72.4% in the conservative group, while all patients in the ablation group had received OAC at baseline. The cumulative 8-year incidences of OAC discontinuation were significantly lower in asymptomatic AF patients than in symptomatic AF patients (47.0% versus 63.1%, log-rank P<0.001), although the difference was not significant after adjustment of the baseline risk factors (HR 1.06, 95% CI 0.79–1.38, P = 0.73) (S1B Fig). The cumulative incidence of OAC discontinuation was significantly lower in asymptomatic AF patients with previous AF-related complications than those without (28.5% and 53.8% at 8-year, log-rank P = 0.003) (S2B Fig). Even after adjustment of the baseline risk factors, the difference remained highly significant (adjusted HR 0.50, 95% CI 0.26–0.90, P = 0.02).

## Discussion

The main findings of this post-hoc subgroup analysis of the previously reported matched AF cohort were the followings: (1) Compared with conservative management, catheter ablation for AF was associated with significantly lower risk for the primary composite outcome of cardiovascular death, HF hospitalization, ischemic stroke or major bleeding in both subgroups of patients with asymptomatic AF and symptomatic AF: (2) Ablation also had significantly lower risk for the primary outcome measure in asymptomatic AF patients with the previous AF-related complications, but not in those without: (3) In the ablation group, the event-free rate from the recurrent atrial tachyarrhythmias was significantly lower in asymptomatic AF patients than in symptomatic AF patients.

Asymptomatic AF is commonly seen in daily clinical practice, and the incidence has been estimated about half of AF patients in several cohort studies [12,13]. Early detection and diagnosis of asymptomatic AF is challenging but important because it leads to a first presentation with acute ischemic stroke or decompensated HF. Furthermore, several studies reported that asymptomatic AF was associated with higher cardiovascular events than symptomatic AF due to higher baseline thromboembolic risk and lower prevalence of OAC use [9,10]. Thus, early diagnosis as well as appropriate management of asymptomatic AF is critically important.

Catheter ablation is a useful strategy to reduce AF burden and improve clinical prognosis of AF patients. The CASTLE-AF (Catheter Ablation versus Standard Conventional Therapy in patients with LEft ventricular dysfunction and Atrial Fibrillation) trial reported that catheter ablation as compared with medical therapy reduced the mortality and HF exacerbation in AF patients with coexisting HF [3]. The CABANA (Catheter ABlation vs ANtiarrhythmic Drug Therapy for Atrial Fibrillation) trial enrolled more general symptomatic AF patients and showed lower risk for the primary composite endpoint of death, disabling stroke, serious bleeding, or cardiac arrest in the ablation group than in the drug therapy group in the on-treatment analysis [4]. Previously, we also reported the superiority of catheter ablation over conservative management for a composite of cardiovascular death, HF hospitalization, ischemic stroke or major bleeding in a real world AF cohort matched with risks including AF duration, AF types, and EHRA symptom grades [8]. In the current study, about 40% of the study

population was asymptomatic AF patients. Catheter ablation was associated with significantly lower risk for the primary outcome measure even in the asymptomatic AF patients. However, the advantage of catheter ablation was significant only in the presence of the previous AF-related complications (prior ischemic stroke or HF hospitalization). The attenuated effect of catheter ablation on reducing the primary outcome in asymptomatic AF without the previous AF-related complications may be explained by the relatively low event rate in the conservative group. In the ablation group, we discontinued OAC after procedure in patients without arrhythmia recurrence according to $CHA_2DS_2$-VASc score and AF type as previously described [14]. Although the OAC discontinuation rate in asymptomatic AF patients was lower than symptomatic AF patients, over 40% of asymptomatic AF patients without previous AF-related complications discontinued OAC at 8-year (S1 Fig). Because detection of AF recurrence in asymptomatic AF patients after ablation is difficult, hasty OAC discontinuation followed by undetected recurrence of asymptomatic AF might increase ischemic stroke after procedure. This indicates OAC discontinuation after ablation in asymptomatic AF patients may be risky and justified only in patients with continuous monitoring of electrocardiography or pulse-wave with implanted cardiac device and/or smart watch.

## Limitations

There were several important limitations in the current study. The first, the most critical limitation was the imbalance of baseline characteristics between the ablation group and the conservative group even after matching. The multivariable analyses might have not adequately eliminated the influence of unmeasured confounders in evaluating the impact of catheter ablation on the primary outcome measure. Second, patients in the conservative group were followed mostly by the referring physicians at the local clinics without a prespecified follow-up protocol, whereas most patients in the ablation group were followed with the recommended follow-up protocol at our hospital or affiliated hospitals, which might have influenced the study results. Third, this post-hoc subgroup analysis may have been underpowered especially in patients with the previous AF-related complications. Finally, patient demographics and risk for clinical outcomes as well as ablation methods in Japan may be different from those outside Japan, so generalizing the results of the present study to populations outside Japan should be done with caution.

## Conclusion

In the post-hoc subgroup analysis of the matched AF cohort, catheter ablation as compared with conservative management was associated with better long-term clinical outcomes even in asymptomatic AF patients, especially in the presence of the previous AF-related complications. Our study warrants future larger studies evaluating the clinical impact of catheter ablation for asymptomatic AF.

## Supporting information

**S1 Fig. Clinical outcomes comparing asymptomatic and symptomatic AF patients in the ablation group.** A) Event free rate from recurrent atrial tachyarrhythmias with a blanking period of 90 days after procedure. B) Discontinuation of OAC. AF = atrial fibrillation; OAC = oral anticoagulation.
(PPTX)

**S2 Fig. Clinical outcomes in asymptomatic AF patients of the ablation group comparing those with and without previous AF-related complications.** A) Event free rate from

recurrent atrial tachyarrhythmias with a blanking period of 90 days after procedure. B) Discontinuation of OAC. AF = atrial fibrillation; OAC = oral anticoagulation.
(PPTX)

**S1 Table. Details of ablation procedure.**
(DOCX)

**S2 Table. Baseline characteristics according to AF symptom grades.**
(DOCX)

**S3 Table. Baseline characteristics of asymptomatic and symptomatic AF patients with and without previous AF-related complications.**
(DOCX)

**S4 Table. Baseline characteristics comparing conservative and ablation groups in asymptomatic AF patients with and without previous AF-related complications.**
(DOCX)

## Acknowledgments

We appreciated all the members of the cardiac catheterization laboratory in Kyoto University Hospital for their contribution to this study.

## Author Contributions

**Conceptualization:** Tetsuma Kawaji.

**Data curation:** Tetsuma Kawaji, Munekazu Tanaka, Shushi Nishiwaki, Takanori Aizawa, Shintaro Yamagami, Akihiro Komasa, Takashi Yoshizawa.

**Investigation:** Tetsuma Kawaji.

**Methodology:** Tetsuma Kawaji, Koh Ono, Takeshi Kimura.

**Project administration:** Satoshi Shizuta.

**Supervision:** Satoshi Shizuta.

**Writing – original draft:** Tetsuma Kawaji.

**Writing – review & editing:** Tetsuma Kawaji, Satoshi Shizuta, Masashi Kato, Takafumi Yokomatsu, Shinji Miki.

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
