## [Decision Letter · Decision Letter 0]

31 Aug 2022

PONE-D-22-17051

Prognostic Impact of Catheter Ablation in Patients with Asymptomatic Atrial Fibrillation

PLOS ONE

Dear Dr. Shizuta,

Thank you for submitting your manuscript to PLOS ONE. After careful consideration, we feel that it has merit but does not fully meet PLOS ONE’s publication criteria as it currently stands. Therefore, we invite you to submit a revised version of the manuscript that addresses the points raised during the review process.

Please address all comments indicated by the Reviewers.

We look forward to receiving your revised manuscript.

Kind regards,

Elena G. Tolkacheva, PhD

Academic Editor

PLOS ONE

Journal Requirements:

2. Thank you for submitting the above manuscript to PLOS ONE. During our internal evaluation of the manuscript, we found significant text overlap between your submission and previous work in the introduction and results. We would like to make you aware that copying extracts from previous publications, especially outside the methods section, word-for-word is unacceptable. In addition, the reproduction of text from published reports has implications for the copyright that may apply to the publications.

Please revise the manuscript to rephrase the duplicated text, cite your sources, and provide details as to how the current manuscript advances on previous work. Please note that further consideration is dependent on the submission of a manuscript that addresses these concerns about the overlap in text with published work.

We will carefully review your manuscript upon resubmission and further consideration of the manuscript is dependent on the text overlap being addressed in full. Please ensure that your revision is thorough as failure to address the concerns to our satisfaction may result in your submission not being considered further.

   "NO"

     "NO"

6. Please include a copy of Table 1 and 2 which you refer to in your text on page 14 and 15.

Additional Editor Comments:

We would ask you also comment on your closely related to previously published work: https://link.springer.com/article/10.1007/s00380-022-02023-0. Please in detail explain how you feel this work differs from your previously published work and why the two manuscripts  represent independent studies.  For further information on our submission guidelines on related manuscripts, please see http://journals.plos.org/plosone/s/submission-guidelines#loc-related-manuscripts.  

Reviewers' comments:

Reviewer's Responses to Questions

**Comments to the Author**

1. Is the manuscript technically sound, and do the data support the conclusions?

Reviewer #1: No

Reviewer #2: Yes

2. Has the statistical analysis been performed appropriately and rigorously? 

Reviewer #1: I Don't Know

Reviewer #2: Yes

3. Have the authors made all data underlying the findings in their manuscript fully available?

Reviewer #1: Yes

Reviewer #2: Yes

4. Is the manuscript presented in an intelligible fashion and written in standard English?

Reviewer #1: Yes

Reviewer #2: Yes

5. Review Comments to the Author

Reviewer #1: The manuscript describes the impact of catheter ablation on clinical outcomes in asymptomatic and symptomatic patients with and without previous AF complications. The manuscript was well written. However, several points need to be re-considered.

1. The major problem with this study was very complicated and confusing. Moreover, the objectives and design of the study did not match. This study aimed to explore the prognostic impact of catheter ablation in asymptomatic AF patients compared with matched AF controls. However, this study included patients with symptomatic AF. The benefits of catheter ablation for symptomatic AF have been well-discussed. If the authors discuss the benefit of the catheter ablation in asymptomatic AF patients, the patients with symptomatic AF should not be included.

2. On page 8, you stated that “the detailed method of the matching was described in the previous report.” However, there was no citation. I don't know whether this matching method is reliable; can't you use a generally accepted method such as propensity score matching?

3. The ablation technique may differ since the study is from a much older period. Please describe the catheter used, mapping method, etc.

4. Further details regarding follow-up methods need to be described.

5. Please describe the ablation results and the medications used in the conservative group.

6. Previous AF complications included the hospitalization exacerbation of HF. These patients are supposed to be symptomatically stabilized by AAD and medications for AF. Can these patients be included in the asymptomatic group?

Reviewer #2: The retrospective study is interesting and reveals that both patients with symptomatic and asymptomatic AF benefit from Af catheter ablation compared to conservative treatment. A limited benefit of ablation therapy is seen in the subset of asymptomatic Af patients who did not have any previous Af-related complications.

In addition to retrospective matched analysis, some limitations need to be mentioned and discussed :

Although, the benefit of AF ablation does not reach statistical significance in the subgroup of asymptomatic Af patients without prior complication, the log-rank p-value is 0.08. A larger prospective study needs to address this issue.

Please provide the clinical baseline for all subgroups: As stated by the authors, the group of patients with asymptomatic AF without previous complications had more often persistent AF and were more frequently of younger age. This may imply that on a longer FU time would be necessary to detect the rare predefined major events in this younger patient group (death, hospitalisation for HF, stroke, haemorrhage). Moreover, as this group is younger, the vent rate is expected to be lower. Therefore, a higher number of patients is needed to detect statistical differences in major event rates, which are are in this young group.

All four patient groups with prior AF-complications are rather small-sized patient groups ranging from 34 to 55 patients (table 2).

6. PLOS authors have the option to publish the peer review history of their article (what does this mean?). If published, this will include your full peer review and any attached files.

Reviewer #1: No

Reviewer #2: **Yes: **Amir JADIDI

---

## [Author Response · Author response to Decision Letter 0]

20 Oct 2022

Response to Reviewers

We deeply appreciate the editors and reviewers for their critically important comments and suggestions on our paper. We have revised our manuscript according to all those comments and suggestions. 

All essential changes in the revised manuscript were highlighted in red font. 

Our replies to the reviewers’ comments and suggestions are written below.

Reviewer #1: The manuscript describes the impact of catheter ablation on clinical outcomes in asymptomatic and symptomatic patients with and without previous AF complications. The manuscript was well written. However, several points need to be re-considered.

1. The major problem with this study was very complicated and confusing. Moreover, the objectives and design of the study did not match. This study aimed to explore the prognostic impact of catheter ablation in asymptomatic AF patients compared with matched AF controls. However, this study included patients with symptomatic AF. The benefits of catheter ablation for symptomatic AF have been well-discussed. If the authors discuss the benefit of the catheter ablation in asymptomatic AF patients, the patients with symptomatic AF should not be included.

Thank you for your important suggestion. As you indicated, the main purpose of the present study was to evaluate the clinical impact of catheter ablation for asymptomatic AF. However, we also think it important to evaluate the impact in the comparison between asymptomatic and symptomatic AF patients. Indeed, in the present study, the benefit of ablation for the primary endpoint was more prominent in symptomatic AF patients than in asymptomatic AF patients. Furthermore, the superiority of ablation over conservative management in asymptomatic AF was significant only in the high-risk subset of patients with previous AF-related complications. 

As you indicated, we admit that our study is a little complicated, so we have tried our best to improve the presentation style in the revised version of the manuscript to avoid confusion. 

Your understanding will be deeply appreciated.

2. On page 8, you stated that “the detailed method of the matching was described in the previous report.” However, there was no citation. I don’t know whether this matching method is reliable; can’t you use a generally accepted method such as propensity score matching?

Thank you for the important comment. Ｗe added descriptions regarding the detailed method for matching between patients receiving catheter ablation and conservative management in the methods section, citing our previous report (Heart Vessels. 2022;37(7):1242-1254). In brief, we first selected a patient in the ablation group and attempted to find a matched control patient based on the prespecified 6 clinical risks; 1) age on the date of first AF documentation; 2) sex; 3) date of first AF documentation; 4) types of AF; 5) EHRA symptom grades (1 to 4); and 6) prior history of HF. 

Although propensity score matching was not used, we made meticulous efforts to find out the appropriate control group of patients with conservative management. However, because we could not perfectly match the patients, there was an imbalance in the prevalence of low body weight, history of malignancy, and CKD between the ablation and conservative groups even after matching. Therefore, we performed multivariate analysis using the Cox proportional hazard model to adjust those imbalances. 

We added descriptions regarding this point in the limitations section, as pasted below.

<Methods>

The current study was a post-hoc subgroup analysis of the previously reported risk-matched study.8 Among 4398 patients with diagnosis of AF in Kyoto University Hospital between January 2005 and March 2015, we performed 1:1 matching to find out an appropriate control patient for a given patient in the ablation group with a prespecified method as follows. First, we selected a patient in the ablation group and attempted to find a matched control patient for the selected patient in the ablation group based on the following clinical information; 1) age on the date of first AF documentation (acceptable range: ± 5 years); 2) sex; 3) date of first AF documentation (acceptable range: ± 365 days); 4) types of AF (paroxysmal or chronic); 5) European Hear Rhythm Association (EHRA) symptom grades (1 to 4); and 6) prior history of HF. When we could not find a matched control patient, the selected patient in the ablation group was excluded from the matched analysis. The control patient who had already been chosen was not matched to another patient in the ablation group to ensure 1:1 matching. Finally, we identified 1074 matched patients (537 patients in the ablation group and 537 patients in the conservative group). The detailed method of the matching was described in the previous report.8

In the present post-hoc subgroup study, we separately assessed the impact of catheter ablation on clinical outcomes in asymptomatic and symptomatic patients with and without previous AF-related complications (ischemic stroke or HF hospitalization).(Line 22 in Page 5 – Line 12 in Page 6)

<Limitations>

The first, the most critical limitation was the imbalance of baseline characteristics between the ablation group and the conservative group even after matching. The multivariable analyses might have not adequately eliminated the influence of unmeasured confounders in evaluating the impact of catheter ablation on the primary outcome measure. (Line 22-26 in Page 13)

3. The ablation technique may differ since the study is from a much older period. Please describe the catheter used, mapping method, etc.

 Thank you for the suggestion. As you indicated, the duration of patient enrollment in the present study was long, 11 years. Actually, the ablation catheter used for PVI changed over time during the period. However, the therapeutic strategy of AF ablation in our center was essentially same during the period, i.e., PVI mostly by RFCA with double circular catheters along with routine tricuspid valve isthmus ablation. Cryoballoon PVI was performed only in 4.4% of patients. 

We added the following descriptions in the methods section and also added S1 Table in the Supplementary materials.

<Methods>

Pulmonary veins isolation (PVI) was performed mostly by radiofrequency catheter ablation using double circular catheters, placing two 20-pollar circular-shaped catheters (Lasso, Biosense Webster or Orbiter PV, C.R. Bard Electrophysiology, Lowell, MA, USA) in ipsilateral superior and inferior pulmonary veins (S1 Table). An 8-mm tip ablation catheter (Fantasista, Japan Lifeline, Tokyo, Japan and NAVISTAER, Biosense Webster, CA, USA) was used from 2004 to 2009, and a 3.5-mm tip irrigation catheter (NAVISTAER THERMOCOOL, Biosense Webster, CA, USA) was used from 2010 to 2015. Cryoballoon (Arctic Front, Medtronic, Inc., MN, USA) was used for PVI only for paroxysmal AF since its introduction to Japan in 2014. Tricuspid valve isthmus ablation was routinely performed regardless of the presence of typical atrial flutter. Superior vena cava was isolated when it was deemed necessary. Complex fractionated atrial electrogram guided ablation was performed when sinus restoration was not obtained after PVI. Additional left atrial linear ablations were performed for sustained atrial tachycardias during the procedure.(Line 24 in Page 6 - Line 10 in Page 7)

<Supplementary material>

S1 Table: Details of ablation procedure

 Ablation group

N=537

Pulmonary vein isolation 537 (100%)

 RFCA 513 (95.6%)

 Cryoballoon 24 (4.4%)

Superior vena cava isolation 144 (26.8%)

Complex fractionated atrial electrogram guided ablation 152 (28.3%)

Tricuspid valve isthmus linear ablation 509 (94.8%)

Mitral valve isthmus linear ablation 14 (2.6%)

Non-pulmonary vein foci ablation 7 (1.3%)

RFCA=radiofrequency catheter ablation.

4. Further details regarding follow-up methods need to be described.

Thank you for your important comment. Patients in the conservative group were followed mostly by the referring physicians at the local clinics without a prespecified follow-up protocol. On the other hand, most patients in the ablation group were followed with the recommended follow-up protocol at our hospital or affiliated hospitals. This difference in the follow-up methods between the groups might have influenced the study results. 

We added the following descriptions in the methods and limitations sections.

<Methods>

Ethics

Follow-up information was obtained by review of hospital-chart and contact by letters and/or phone-call to the patient, relatives, and/or referring physicians. The follow-up protocol in the ablation groups was described in the next section. On the other hand, there were no prespecified follow-up protocol in the conservative group because the present study was a retrospective analysis. The study protocol was approved by the institutional review board in Kyoto University Hospital. Written informed consent for the catheter ablation procedure and follow-up was obtained from all patients in ablation group and we got consent for the enrollment to the study from all patients in the conservative group at the time of follow-up contact. (Line 13 - 22 in Page 6)

<Methods>

A 12-lead electrocardiogram was routinely measured at each clinical visit and 24-hour Holter monitoring was recommended at 3-, 6-, 12-month and yearly thereafter. Antiarrhythmic drug was discontinued before ablation procedure, and was restarted only when recurrent atrial tachyarrhythmias were detected. The second ablation was recommended to patients with recurrent atrial tachyarrhythmias after the blanking period of 3 months. Oral anticoagulant (OAC) was continued for at least 3 months after procedure. Thereafter, discontinuation of OAC in patients without arrhythmia recurrence was left to the discretion of the attending physician. (Line 11 - 17 in Page 7)

<Limitations>

Second, patients in the conservative group were followed mostly by the referring physicians at the local clinics without a prespecified follow-up protocol, whereas most patients in the ablation group were followed with the recommended follow-up protocol at our hospital or affiliated hospitals, which might have influenced the study results. (Line 26 in Page 13- 4 in Page 14)

5. Please describe the ablation results and the medications used in the conservative group.

Thank you for the important comment. According to your suggestion, we added the following Tables and Figures showing the details of ablation results and medications, as pasted below.

S1 Table: Details of ablation procedure

 Ablation group

N=537

Pulmonary vein isolation 537 (100%)

 RFCA 513 (95.6%)

 Cryoballoon 24 (4.4%)

Superior vena cava isolation 144 (26.8%)

Complex fractionated atrial electrogram guided ablation 152 (28.3%)

Tricuspid valve isthmus linear ablation 509 (94.8%)

Mitral valve isthmus linear ablation 14 (2.6%)

Non-pulmonary vein foci ablation 7 (1.3%)

RFCA=radiofrequency catheter ablation.

Table 1: Baseline characteristics comparing conservative versus ablation groups in asymptomatic and symptomatic AF patients. 

Note: Only the information regarding medications at baseline are shown here

Baseline characteristics Asymptomatic Symptomatic

 Conservative

group

N=207 Ablation

group

N=207 P value Conservative

group

N=330 Ablation

group

N=330 P value

Medications 

 Oral anticoagulant 169 (81.6%) 207 (100%) <0.001 219 (66.4%) 330 (100%) <0.001

 Warfarin 104 (50.2%) 104 (50.2%) 1.00 109 (33.0%) 175 (53.0%) <0.001

DOACs 65 (31.4%) 103 (49.8%) <0.001 112 (33.9%) 155 (47.0%) <0.001

 Antiplatelets 50 (24.2%) 43 (20.8%) 0.41 75 (22.7%) 55 (16.7%) 0.049

 Antiarrhythmic drugs 31 (15.0%) 45 (21.7%) 0.07 120 (36.4%) 118 (35.8%) 0.87

 Beta blockers 82 (39.6%) 63 (30.4%) 0.05 107 (32.4%) 107 (32.4%) 1.00

 Verapamil/diltiazem 34 (16.4%) 18 (8.4%) 0.02 67 (20.3%) 38 (11.5%) 0.002

 Digitalis 30 (14.5%) 21 (10.1%) 0.18 32 (9.7%) 26 (7.9%) 0.41

 ACEI/ARB 91 (44.0%) 100 (48.3%) 0.37 117 (35.5%) 138 (41.8%) 0.09

S1 Figure: Clinical outcomes comparing asymptomatic and symptomatic AF patients in the ablation group 

A) Event free rate from recurrent atrial tachyarrhythmias with a blanking period of 90 days after procedure

B) Discontinuation of OAC

S2 Figure: Clinical outcomes in asymptomatic AF patients of the ablation group comparing those with and without the previous AF complications

A) Event free rate from recurrent atrial tachyarrhythmias with a blanking period of 90 days after procedure

B) Discontinuation of OAC

Previous AF complications included the hospitalization exacerbation of HF. These patients are supposed to be symptomatically stabilized by AAD and medications for AF. Can these patients be included in the asymptomatic group?

Thank you for your important comment. As you indicated, it may be controversial to categorize patients without any AF-related symptoms at the enrollment of the study as asymptomatic when a history of HF hospitalization was present. However, in the EHRA report (Europace 2007;9:1006–23), AF-related symptoms are recommended to be assessed at the baseline (patient enrollment). Also, importantly, we graded the symptom status at the start of the previous study (Heart Vessels. 2022;37(7):1242-1254), because symptom status was one of the prespecified 6 clinical risk factors for the matching, as written previously at the bottom of Page 2 of this reply document. 

Your understanding will be deeply appreciated

Reviewer #2: The retrospective study is interesting and reveals that both patients with symptomatic and asymptomatic AF benefit from Af catheter ablation compared to conservative treatment. A limited benefit of ablation therapy is seen in the subset of asymptomatic Af patients who did not have any previous Af-related complications.

 Thank you for your important comment on our paper. Our replies to your comments are written below.

In addition to retrospective matched analysis, some limitations need to be mentioned and discussed :

Although, the benefit of AF ablation does not reach statistical significance in the subgroup of asymptomatic Af patients without prior complication, the log-rank p-value is 0.08. A larger prospective study needs to address this issue.

Thank you for your valuable comments. As you indicated, the benefit of catheter ablation in asymptomatic AF patients without previous AF complications was with marginal insignificance, presumably due to the relatively small number of patients. 

We added the following sentences in the limitations section.

<Limitations>

Third, this post-hoc subgroup analysis may have been underpowered especially in patients with the previous AF-related complications. (Line 4 - 5 in Page 14)

Please provide the clinical baseline for all subgroups: As stated by the authors, the group of patients with asymptomatic AF without previous complications had more often persistent AF and were more frequently of younger age. This may imply that on a longer FU time would be necessary to detect the rare predefined major events in this younger patient group (death, hospitalisation for HF, stroke, haemorrhage). Moreover, as this group is younger, the vent rate is expected to be lower. Therefore, a higher number of patients is needed to detect statistical differences in major event rates, which are are in this young group.

Thank you for your important comment. As you indicated, patients with previous AF complications had significantly higher cardiovascular risks and lower cardiac function relative to those without, regardless of AF symptoms. The differences in the baseline characteristics among 4 subgroups may have influenced the study results. However, the baseline characteristics of the ablation versus conservative groups were generally well balanced in all the subgroups. 

According to your suggestion, we added S3 and S4 Tables, as pasted below. We also added descriptions regarding this point in the results and the limitations sections.

S3 Table: Baseline characteristics of asymptomatic and symptomatic AF patients with and without previous AF-related complications

 Asymptomatic 

 No previous

AF-related complications

N=317 Asymptomatic 

 With　previous

AF-related complications

N=97 Symptomatic 

 No　previous

AF-related complications

N=584 Symptomatic 

 With previous

AF-related complications

N=76 P value

Age (years old) 65.8±6.4 67.4±7.7 66.0±8.8 69.2±7.6 <0.001

 ≥ 75 years old 27 (8.5%) 14 (14.4%) 94 (16.1%) 21 (27.6%) <0.001

Women 49 (15.5%) 15 (15.5%) 199 (34.1%) 25 (32.9%) <0.001

Weight (kg) 65.7±11.8 62.4±13.3 62.1±12.7 ｗ <0.001

 Low body weight 43 (14.3%) 28 (30.1%) 125 (22.8%) 23 (30.7%) <0.001

Non-paroxysmal AF 168 (53.0%) 64 (66.0%) 90 (15.4%) 14 (18.4%) <0.001

AF duration (years) 1.2 (0.4-4.3) 1.5 (0.5-5.3) 1.4 (0.4-3.7) 1.7 (0.5-4.7) 0.34

Hypertension 212 (66.9%) 71 (73.2%) 356 (61.0%) 53 (69.7%) 0.04

Diabetes 61 (19.2%) 22 (22.7%) 99 (17.0%) 16 (21.1%) 0.49

Previous AF-related complications 0 (0.0%) 97 (100%) 0 (0.0%) 76 (100%) <0.001

 History of heart failure hospitalization 0 (0.0%) 37 (38.1%) 0 (0.0%) 76 (100%) <0.001

 Ischemic stroke 0 (0.0%) 67 (69.1%) 0 (0.0%) 60 (79.0%) <0.001

CHA2DS2-VASc score 1.8±1.2 3.9±1.3 2.0±1.3 4.2±1.4 <0.001

 ≥ 2 172 (54.3%) 95 (97.9%) 364 (62.3%) 75 (98.7%) <0.002

History of malignancy 317 (18.6%) 14 (14.4%) 98 (16.8%) 12 (15.8%) 0.77

eGFR (ml/min/1.73m2) 52.2±16.9 48.5±14.0 55.2±17.3 47.4±14.9 <0.001

 ≤60 ml/min/1.73m2 230 (74.7%) 79 (83.2%) 364 (65.0%) 55 (75.3%) <0.001

Echocardiographic data 

 Left ventricular ejection fraction (%) 63.1±11.0 59.1±15.9 66.4±10.0 64.3±12.3 <0.001

 ≤ 40 % 11 (3.9%) 15 (16.1%) 9 (1.6%) 4 (5.5%) <0.001

 Left atrial diameter (mm) 43.0±8.1 44.5±8.8 39.5±7.3 40.0±7.6 <0.001

 ≥ 50 mm 45 (16.0%) 23 (25.0%) 53 (9.6%) 7 (9.6%) <0.001

Medications 

 Oral anticoagulat 285 (89.9) 91 (93.8%) 481 (82.4%) 68 (89.5%) <0.001

 Warfarin 143 (45.1%) 65 (67.0%) 249 (42.6%) 35 (46.1%) <0.001

Direct oral anticoagulants 142 (44.8%) 26 (26.8%) 234 (40.1%) 33 (43.4%) 0.01

 Antiplatelet use 59 (18.6%) 34 (35.1%) 106 (18.2%) 24 (31.6%) <0.001

 Anti-arrhythmic drugs 56 (17.7%) 20 (20.6%) 207 (35.5%) 31 (40.8%) <0.001

 Beta blockers 111 (35.0%) 34 (35.1%) 183 (31.3%) 31 (40.8%) 0.33

 Verapamil/diltiazem 42 (13.3%) 10 (10.3%) 92 (15.8%) 76 (17.1%) 0.39

 Digitalis 35 (11.0%) 16 (16.5%) 50 (8.6%) 8 (10.5%) 0.13

 ACEI/ARB 142 (44.8%) 49 (50.5%) 222 (38.0%) 33 (43.4%) 0.05

S4 Table: Baseline characteristics comparing conservative and ablation groups in asymptomatic AF patients with and without previous AF-related complications

Baseline characteristics Previous AF-related complications No Previous AF-related complications

 Conservative

group

N=42 Ablation

group

N=55 P value Conservative

group

N=165 Ablation

group

N=152 P value

Age (years old) 67.5±8.4 67.4±7.3 0.96 66.0±6.5 65.6±6.3 0.53

 ≥ 75 years old 7 (16.7%) 7 (12.7%) 0.59 14 (8.5%) 13 (8.6%) 0.98

Women 5 (11.9%) 10 (18.2%) 0.39 27 (16.4%) 22 (14.5%) 0.64

Weight (kg) 61.1±11.6 63.2±14.4 0.46 63.6±12.3 67.8±10.9 0.002

 Low body weight 13 (34.2%) 15 (27.3%) 0.47 31 (21.0%) 12 (7.9%) 0.001

Non-paroxysmal AF 26 (61.9%) 38 (69.1%) 0.46 90 (54.5%) 78 (51.3%) 0.56

AF duration (years) 1.3 (0.5-4.4) 2.1 (0.5-5.8) 0.20 1.3 (0.4-4.6) 1.0 (0.4-3.8) 0.43

Hypertension 31 (73.8%) 40 (72.7%) 0.91 103 (62.4%) 109 (71.7%) 0.08

Diabetes 11 (26.2%) 11 (20.0%) 0.47 38 (23.0%) 23 (15.1%) 0.07

Previous AF-related complications 42 (100%) 55 (100%) - 0 (0%) 0 (0%) -

 History of heart failure hospitalization 19 (45.2%) 18 (32.7%) 0.21 0 (0%) 0 (0%) -

 Ischemic stroke 25 (59.5%) 42 (76.4%) 0.08 0 (0%) 0 (0%) -

CHA2DS2-VASc score 3.8±1.3 3.9±1.2 0.65 1.8±1.2 1.7±1.1 0.63

 ≥ 2 40 (95.2%) 55 (100%) 0.18 93 (56.4%) 79 (52.0%) 0.43

History of malignancy 7 (16.7%) 7 (12.7%) 0.59 46 (27.9%) 13 (8.6%) <0.001

eGFR (ml/min/1.73m2) 48.7±15.5 48.3±12.9 0.89 51.6±18.9 52.9±14.6 0.50

 ≤60 ml/min/1.73m2 31 (77.5%) 48 (87.3%) 0.21 109 (69.9%) 121 (79.6%) 0.049

Echocardiographic data 

 Left ventricular ejection fraction (%) 55.0±16.4 62.0±15.1 0.04 64.6±11.2 61.8±10.7 0.03

 ≤ 40 % 8 (21.1%) 7 (12.7%) 0.29 2 (1.5%) 9 (5.9%) 0.046

 Left atrial diameter (mm) 45.8±10.3 43.6±7.6 0.24 43.9±9.3 42.3±6.7 0.09

 ≥ 50 mm 11 (29.7%) 12 (21.8%) 0.39 26 (20.0%) 19 (12.5%) 0.09

Medications 

 Oral anticoagulat 36 (85.6%) 55 (100%) 0.001 133 (80.6%) 152 (100%) <0.001

 Warfarin 25 (59.5%) 40 (72.7%) 0.17 79 (47.9%) 64 (42.1%) 0.30

Direct oral anticoagulants 11 (26.2%) 15 (27.3%) 0.91 54 (32.7%) 88 (57.9%) <0.001

 Antiplatelet use 14 (33.3%) 20 (36.4%) 0.76 36 (21.8%) 23 (15.1%) 0.12

 Anti-arrhythmic drugs 5 (11.9%) 15 (27.3%) 0.06 26 (15.8%) 30 (19.7%) 0.35

 Beta blockers 18 (42.9%) 16 (29.1%) 0.16 64 (38.8%) 47 (30.9%) 0.14

 Verapamil/diltiazem 5 (11.9%) 5 (9.1%) 0.65 29 (17.6%) 13 (8.6%) 0.02

 Digitalis 5 (11.9%) 11 (20.0%) 0.28 25 (15.2%) 10 (6.6%) 0.01

 ACEI/ARB 19 (45.2%) 30 (54.6%) 0.36 72 (43.6%) 70 (46.1%) 0.67

<Methods>

Patients with the previous AF-related complications had significantly higher cardiovascular risks and lower cardiac function than those without, regardless of AF symptoms (S3 Table). (Line 3 - 4 in Page 10)

<Limitations>

he first, the most critical limitation was the imbalance of baseline characteristics between the ablation group and the conservative group even after matching. The multivariable analyses might have not adequately eliminated the influence of unmeasured confounders in evaluating the impact of catheter ablation on the primary outcome measure. (Line 22 – 26 in Page 13)

All four patient groups with prior AF-complications are rather small-sized patient groups ranging from 34 to 55 patients (table 2).

Thank you for your important comment. As you indicated, the number of patients in the subgroups with previous AF complications were relatively small, as shown in S3 and S4 Tables shown above.

We added descriptions regarding this point in the limitations section.

Third, this post-hoc subgroup analysis may have been underpowered especially in patients with the previous AF-related complications. (Line 4 - 5 in Page 14)

---

## [Decision Letter · Decision Letter 1]

2 Dec 2022

Prognostic Impact of Catheter Ablation in Patients with Asymptomatic Atrial Fibrillation

PONE-D-22-17051R1

Dear Dr. Shizuta,

We’re pleased to inform you that your manuscript has been judged scientifically suitable for publication and will be formally accepted for publication once it meets all outstanding technical requirements.

Kind regards,

Elena G. Tolkacheva, PhD

Academic Editor

PLOS ONE

Additional Editor Comments (optional):

Reviewers' comments:

Reviewer's Responses to Questions

**Comments to the Author**

1. If the authors have adequately addressed your comments raised in a previous round of review and you feel that this manuscript is now acceptable for publication, you may indicate that here to bypass the “Comments to the Author” section, enter your conflict of interest statement in the “Confidential to Editor” section, and submit your "Accept" recommendation.

Reviewer #1: All comments have been addressed

Reviewer #2: All comments have been addressed

2. Is the manuscript technically sound, and do the data support the conclusions?

Reviewer #1: Yes

Reviewer #2: Yes

3. Has the statistical analysis been performed appropriately and rigorously? 

Reviewer #1: I Don't Know

Reviewer #2: Yes

4. Have the authors made all data underlying the findings in their manuscript fully available?

Reviewer #1: Yes

Reviewer #2: Yes

5. Is the manuscript presented in an intelligible fashion and written in standard English?

Reviewer #1: Yes

Reviewer #2: Yes

6. Review Comments to the Author

Reviewer #1: All reviewer comments are appropriately revised and addressed.

I have no more additional questions about this manuscript.

Reviewer #2: The authors have adequately adressed my comments, and mention the limitations of this rtrospective analysis.

7. PLOS authors have the option to publish the peer review history of their article (what does this mean?). If published, this will include your full peer review and any attached files.

Reviewer #1: No

Reviewer #2: **Yes: **Amir JADIDI

---

## [Editor Report · Acceptance letter]

6 Dec 2022

PONE-D-22-17051R1 

Prognostic Impact of Catheter Ablation in Patients with Asymptomatic Atrial Fibrillation 

Dear Dr. Shizuta:

I'm pleased to inform you that your manuscript has been deemed suitable for publication in PLOS ONE. Congratulations! Your manuscript is now with our production department. 

Kind regards, 

on behalf of

Dr. Elena G. Tolkacheva 

Academic Editor

PLOS ONE